# Platinum oxide formation under oxygen evolution reaction conditions

Leon Jacobse [1,6] ✉, Ralf Schuster[2], Mona Kohantorabi [1],
Daniel Silvan Dolling[1,3], Johannes Pfrommer[1,4], Xin Deng [1,2], Tim Weber[4],
Olof Gutowski[5], Ann-Christin Dippel [5], Olaf Brummel [2], Yaroslava Lykhach [2],
Heshmat Noei [1], Herbert Over [4], Jörg Libuda [2], Vedran Vonk [1] &
Andreas Stierle [1,3]

Electrocatalyst degradation, often caused by oxidative processes, forms a large barrier for the wide-spread application of electrolysers and fuel cells, which are crucial for a sustainable energy society. A detailed understanding of the catalyst surface structure under oxygen evolution reaction (OER) conditions is, therefore, required to design more stable catalysts. Here, we study the oxidation of a Pt(111) model electrode under operando conditions combining High-Energy Surface X-ray Diffraction (HE-SXRD) with a Rotating Disk Electrode (RDE) in a unique experimental setup. This approach allows us to follow the atomic structure of the electrode-electrolyte interface under oxygen evolution reaction conditions under hitherto unexplored potential regimes. We find that the Pt(111) surface gets electro-oxidized in a layer-by-layer fashion. From ex situ X-ray Reflectivity (XRR) and X-ray Photoelectron Spectroscopy (XPS) measurements we find that a sub-nm thick, $PtO_2$ oxide film is forming, which deactivates the surface and leads to surface roughening. Our results provide important insights into the electrochemical oxidation of platinum electrocatalysts and resolves crucial differences to thermal oxidation processes.

Platinum is the electrocatalyst material of choice in many electrochemical energy conversion devices, due to its high activity for water splitting in electrolyzers and the reverse reactions in hydrogen fuel cells[1,2]. A widespread commercial application of fuel cells and electrolyzers is hindered by the ongoing degradation of the expensive electrode material during operation. Because of the intermittance of renewable energies, fuel cells cannot be operated under steady state conditions. Unfortunately, switching on and off a fuel cell is one of the most important reasons for its degradation[3]. Such degradation processes include faceting, dissolution, and roughening of the electrocatalyst surface and support, and occur mainly upon exposure to oxidizing conditions or, in the case of catalyst dissolution, during the subsequent reduction[4,5]. The platinum-electrolyte interface is one of the best-controlled electrochemical interfaces, allowing reproducible experimental studies, also accessible for theoretical calculations[6]. Insights obtained for platinum model electrodes contribute to the development of new electrocatalysts and operating protocols that minimize electrode degradation. However, despite the detailed studies of the platinum-electrolyte interface, there is a lack of fundamental understanding of processes at higher potentials in the

[1]Centre for X-ray and Nano Science CXNS, Deutsches Elektronen-Synchrotron DESY, Hamburg, Germany. [2]Interface Research and Catalysis, Erlangen Center for Interface Research and Catalysis, Friedrich-Alexander-Universität Erlangen-Nürnberg, Erlangen, Germany. [3]Fachbereich Physik, Universität Hamburg, Hamburg, Germany. [4]Institute of Physical Chemistry and Center for Materials Research, Justus Liebig Universität Gießen, Gießen, Germany. [5]Deutsches Elektronen-Synchrotron DESY, Hamburg, Germany. [6]Present address: Department of Interface Science, Fritz Haber Institute of the Max Planck Society, Berlin, Germany. ✉e-mail: jacobse@fhi-berlin.mpg.de

surface oxidation and oxygen evolution reaction (OER) regime. Understanding the Pt catalyst dynamics under such conditions is relevant for proton exchange membrane fuel cells, as during shutdown, the formation of an air/fuel boundary at the anode can expose the cathode to potentials up to 2.0 V[3,7]. The mechanism for the formation and the atomic structure of platinum oxides are of special current interest as the involved structural rearrangement is playing a key role in the overall degradation process[8–10].

Surface sensitive X-ray diffraction (SXRD) has proven to be a very powerful tool to study the restructuring of Pt surfaces under electrochemical conditions[8–19]. These experiments showed that the initial oxidation of Pt(111), starting around 1.0 V vs. the reversible hydrogen electrode (RHE), occurs via a so-called place-exchange (PE) process. Here, Pt atoms are extracted perpendicular from the topmost atomic layer by around a single atomic layer distance, which is facilitated by the presence of oxygen. Simultaneously, however, there is a second process occurring where atoms are displaced to positions that are no longer in registry with the underlying lattice, during which Pt oxide is formed[9]. As the PE process is observed to be reversible, it is the formation of this disordered oxide that leads to the surface roughening under potential cycling conditions[5,13,19–21]. Although mild surface roughening can be applied to generate surface sites that exhibit a high electrocatalytic activity[22], it ultimately leads to irreversible performance degradation due to the loss of Pt into the electrolyte[23–25]. On a fundamental level, density functional theory (DFT) calculations have provided important insights in the PE process, e.g., regarding the position of the oxygen atoms, the stability of ordered oxide chains/stripes, and the role of the electrolyte[8,26–29]. However, there is a lack of theoretical and experimental information on the evolution of the Pt oxide growth beyond the monolayer regime. Under these conditions, at potentials >1.5 V, the PE description for the oxidation process is expected to break down. Interestingly, it is in this same potential regime where Pt exhibits a significant activity for the OER. Kinetic Monte Carlo simulations of extensive thermal oxidation were performed, but it is unclear to what extent these results can be translated to electrochemical conditions[30].

Recently, we have developed an instrument combining HE-SXRD and rotating disk electrode (RDE) experiments, enabling controlled mass-transport conditions for measurements towards industrially relevant operando conditions, especially under strong gas evolution leading to an uncontrolled potential at the surface[31]. Here, we use this instrument to study the stability of the Pt(111) surface in a wide potential window, including both the oxygen reduction reaction (ORR) and the OER up to a potential of 2.09 V. We demonstrate that under OER conditions, a platinum oxide with a thickness exceeding a single monolayer (ML) forms at the surface. This oxide turns out to be strongly disordered, since no oxide diffraction signal can be recorded within the resolution of our experiment. We have therefore performed complementary ex situ X-ray reflectivity (XRR) and X-ray photoelectron spectroscopy (XPS) experiments to study the thickness, density, and chemical nature of the oxide layer itself. The experiments resolve that the platinum surface is dissolved in a layer-by-layer fashion and that the thickness of the formed $PtO_2$ layer increases linearly with the applied electrochemical potential. Our results provide detailed atomistic insights in the chemical and structural properties of Pt(111) under ORR and OER conditions, which are crucial for the development of catalysts exhibiting higher activity and stability not only for these reactions but for all reactions involving oxidizing conditions.

## Results and discussion

To resolve the structure of Pt(111) at oxidizing potentials and ultimately during the OER, we performed RDE-SXRD measurements starting from the reduced surface up to 2.09 V, generating a current density of 6 mA cm$^{-2}$ (see Fig. S2). Previous experiments, employing static conditions, were limited to significantly lower potentials (<1.6 V)

and steady-state current densities that were three orders of magnitude lower[9,13]. To avoid effects originating from the interaction between (both anionic and cationic) electrolyte species, we use a 0.1 M $HClO_4$ electrolyte solution[32,33]. At potentials below 0.9 V, Pt(111) demonstrates significant ORR activity, but this does not lead to changes in the crystal truncation rod (CTR) profiles (see Figs. S2 and S3). This confirms previous results obtained under static conditions, indicating that the presence of molecular oxygen does not have a significant effect on the electro-oxidation process[34].

Figure 1 shows the experimental CTR profiles as a function of the applied potential up to 2.09 V together with the respective fits. In between the measurements, the potential was increased stepwise and held for at least 10 min for the surface to equilibrate. Repetitive measurements, as well as previous measurements[9,35], indicate a mostly static surface structure after this waiting time. The measurements shown here were performed in $O_2$-saturated electrolyte, but the oxidation process is not significantly affected by the presence of $O_2$ (see Figs. S4 and S5). Drastic changes are observed once potentials above 1 V are applied. The intensity in the CTR minima is decreasing, which is an indication for an increasing number of Pt vacancies in the first atomic layer„ related to extracted Pt atoms, as discussed in more detail below[36]. Surprisingly, when the potential is further increased, the intensity in the CTR minima is increasing again, which indicates a subsequent smoothening of the electrode-electrolyte interface. The evolution of the measured intensity at the surface sensitive (1, 1, 1.5) reflection as a function of potential shows, in fact, an oscillatory behavior as a function of the applied potential (see Fig. S5) with intensity minima (i.e., maximum roughness) around 1.25 and 2.09 V. This is very similar to the observation of a layer-by-layer epitaxial growth resulting in growth oscillations as a function of deposition time[36], only that here we have to deal with the reverse process, i.e., a layer-by-layer extraction of Pt atoms from the metallic surface. Note that, in our experiment, the process is driven by the applied potential, since at constant potential, the oxidation kinetics are very slow compared to the timescale of the experiment. In case of a continued (kinetic) roughening of the interface, a continuous decrease in the CTR intensity would be expected for higher potentials[9].

Next, we will discuss the quantitative analysis of the CTR data. We used a model as illustrated in Fig. 2A, where a fraction of the Pt atoms of the topmost atomic layer of the Pt (111) electrode ($Pt_{PE}$, orange) is allowed to move perpendicular to the surface layer ($Pt_{sur}$, blue). The oxygen atoms are not included in the fits as their scattering contribution compared to Pt is negligible. To account for the formation of a disordered oxide, which does not contribute to the CTRs because the atoms lose their registry to the Pt(111) surface, mass-conservation between the different Pt sites was lifted[9]. Occupation probabilities (occupancies) and displacements were fitted for each of the three surface atom types involved in the oxidation process, i.e., $Pt_{PE}$, $Pt_{sur}$, and the layer directly subsurface ($Pt_{base}$, gray in Fig. 2A). Isotropic root mean square displacements $u$ (static Debye–Waller factors, $B = 8\pi u^2$) were fitted for all three layers. Online ICP-MS experiments[23,33,37] have demonstrated that under our applied oxidizing conditions, the amount of dissolved Pt is far below the sensitivity of the SXRD experiment (<< 1% of a ML), and can therefore be neglected. This enables us to use the net reduction in occupancies of the observable layers ($Pt_{base}$, $Pt_{sur}$, $Pt_{PE}$) to determine (the upper limit for) the amount of disordered oxide ($Pt_{ox}$). Importantly, the CTR signal is only sensitive to periodic structures at the metal-oxide interface, which exhibit the same (or integer multiple) in-plane lattice constant as the Pt(111) support.

The fit results are shown as the colored lines in Fig. 1 and the corresponding parameter values in Fig. 2B–D. From these results, we can identify four different potential regimes, in which we observe a different oxidation behavior. For the initial oxidation in the potential regime of 1–1.2 V, up to 0.2 ML $Pt_{PE}$ atoms are formed, residing ~2 Å above their initial surface position. Their displacement distribution $u$ is

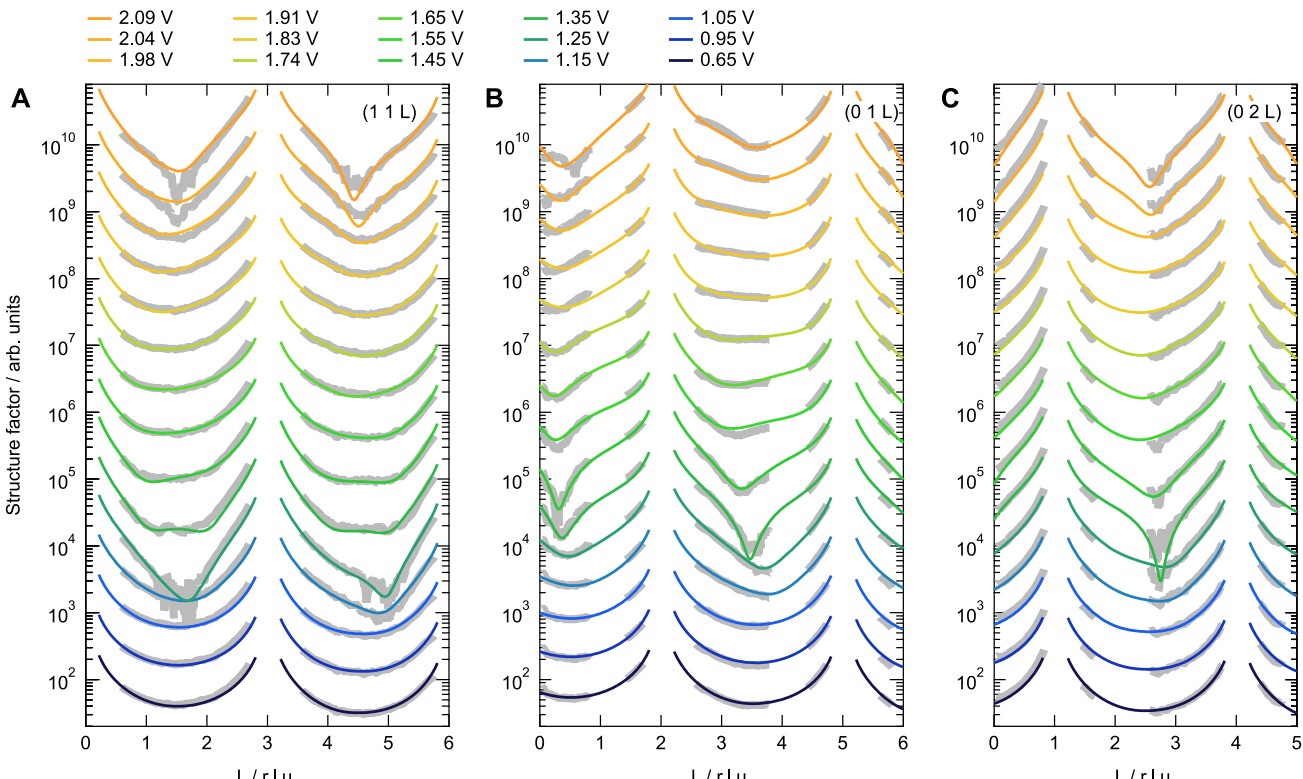

**Fig. 1 | CTR data under oxidizing conditions.** (1 1), (0 1), and (0 2) CTRs (gray) and fits (color) as a function of potential (**A**–**C**, respectively) measured in $O_2$-saturated 0.1 M $HClO_4$ electrolyte. The fit parameter values are shown in Fig. 2. Potentials are corrected for the full ohmic resistance of the electrolyte (150 ± 10 Ω, see "Methods"). The data are vertically offset for clarity. Source data are provided as a Source data file.

small, indicating a well-defined geometry of the $Pt_{PE}$ atoms (see Fig. 2D). In this potential regime, roughly half of the reduction in the $Pt_{sur}$ occupancy is compensated by the increasing $Pt_{PE}$ occupancy. If the oxidation is performed using a slow potential sweep (open symbols in Fig. 2B[9]) rather than stepwise (here), the conversion of $Pt_{sur}$ into $Pt_{PE}$ is almost completely mass-conserved. The remaining $Pt_{sur}$ atoms undergo a slight inward relaxation, and their $u$ values are characteristic for the Pt(111) surface.

In the potential regime between 1.2 and 1.6 V, a different behavior is observed: the occupancy of the $Pt_{PE}$ atoms remains at 0.2 ML, but they undergo an outward relaxation. We propose that originates from a stronger coordination of the $Pt_{PE}$ atoms by oxygen, leading to a weakening of the bonds to the $Pt_{sur}$ layer. In addition, $u$ is strongly increasing, caused by heterogeneous displacements of the Pt atoms on PE sites. Despite the constant $Pt_{PE}$ occupancy, the occupation of the $Pt_{sur}$ sites continues to decrease until this layer can no longer be observed, evidencing significant oxide formation as dissolution is insignificant[23]. This is only to a very small extent compensated by the appearance of a $Pt_{PE}$ species that is displaced to a distinctly different height (see below). Since no additional diffraction signal from ordered structures with a different periodicity than the substrate can be observed over the whole potential range (see Fig. S6), we conclude that the formed oxide is highly disordered with an even smaller domain size as compared to a few nm for thermal oxidation at elevated temperatures[38,39]. Based on an analysis of the integrated intensity of the CTR signal, we estimate that the signal of two O-Pt-O monolayer high oxide domains with a lateral size of around 3 nm is at the detection limit of our experiment.

In the third potential regime between 1.60 and 2 V, a transition occurs where the oxidation process leads to a different structure, as can be recognized from several changes in the fit parameter values. First, the last remaining $Pt_{sur}$ atoms become strongly disordered (large

$u$ values, see Fig. 2D) and show a decrease in occupancy (Fig. 2), compatible with a complete transition of a metal- to oxide-terminated surface. This means that the $Pt_{PE}$ atoms now sit on three-fold hollow sites with respect to the $Pt_{base}$ layer, whereas the latter now forms the topmost metallic Pt layer. Consequently, around 0.2 ML of Pt atoms in the growing oxide reside on a well-defined height in three-fold registry with the $Pt_{base}$ layer. This height, however, is different from the height where the $Pt_{PE}$ atoms resided previously, namely around 3.25 ± 0.03 Å with respect to the $Pt_{base}$ layer. Similar as observed at lower potentials, we see that the $Pt_{PE}$ layer becomes somewhat more disordered with increasing potential. Considering this trend, as well as the observed outward relaxations, it seems logical that the $Pt_{PE}$ layer is formed from the $Pt_{sur}$ atoms (with possible contributions from $Pt_{ox}$) rather than a "collapse" of the topmost layer. However, note that our experiment gives no direct insight in the displacement pathways of the individual atoms.

We find a distance of 3.25 ± 0.03 Å between the $Pt_{PE}$ atoms and the topmost remaining Pt layer ($Pt_{base}$). This value is significantly larger than the 2.1 Å reported for $\alpha$-$PtO_2$ oxide islands, which are separated from the metallic Pt surface by only a single layer of oxygen atoms[40], suggesting the presence of adsorbed oxygen at the $Pt_{base}$ layer. In comparison, bulk $\alpha$-$PtO_2$ crystallizes in a hexagonal phase with $a = b = 3.1$ Å and a distance $c$ between subsequently stacked, hexagonal O-Pt-O trilayers of 4.16 Å (see inset in Fig. 2A)[41]. However, previously it was found that the out-of-plane lattice constant becomes drastically smaller for thin $\alpha$-$PtO_2$ layers (i.e., 3.62 Å for two O-Pt-O trilayers formed by thermal oxidation on Pt(111)[38]). We propose that the Pt-Pt interlayer distance reduction should be ascribed to enhanced van der Waals interactions between the charge-neutral O-Pt-O trilayer in the oxide and the oxygen-covered Pt substrate surface. In this situation, the $Pt_{base}$ layer could be argued to be a PtO interlayer, an assignment previously used by Baroody et al.[42].

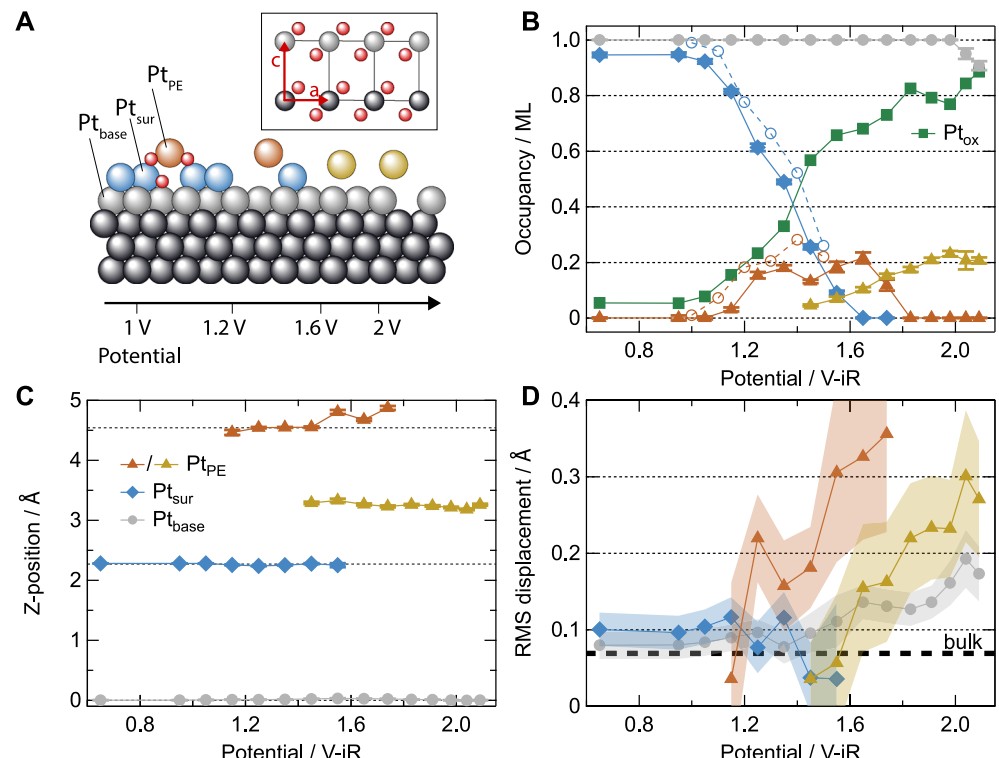

**Fig. 2 | Model and fit parameter values. A** Model depiction of the PE process and subsequent surface evolution. The inset shows the bulk $\alpha$-$PtO_2$ structure. Occupancies, displacements, root mean square displacement distributions (static Debye–Waller factors) of the $Pt_{sur}$ (blue diamonds), $Pt_{PE}$ (orange/yellow triangles), and $Pt_{base}$ (gray circles) layers as a function of sample potential (**B**–**D**, respectively). The occupancy of the disordered $Pt_{ox}$ species (green squares) is calculated from the other occupancies and the absence of significant dissolution under these conditions[23]. Open symbols in (**B**) are data from the slow oxidation sweep experiment described in ref. [9]. The dashed lines in (**C**) indicate the bulk Pt(111) layer spacing. The bold dashed line in (**D**) indicates the Debye–Waller factor of bulk Pt at room temperature[65]. Error bars (only significantly larger than the data markers for the DW factors) indicate the errors of the least-squares fit, whereas the real uncertainties may be larger. Potentials are corrected for the full ohmic resistance of the electrolyte ($150 \pm 10\,\Omega$, see "Methods"). Source data are provided as a Source data file.

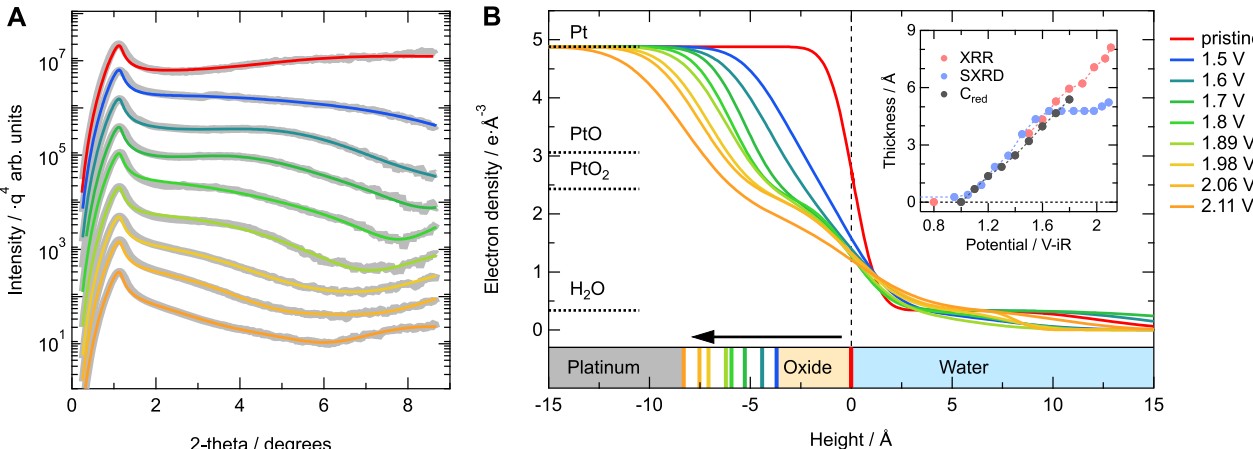

**Fig. 3 | X-ray reflectivity results. A** X-ray reflectivity data (gray) and the corresponding fits (colored lines) showing the changes due to the formation of an oxide layer. For clarity, the data are divided by $q^{-4}$ and vertically offset. **B** Electron density profiles determined from the XRR fit results, indicating an increase in both the oxide thickness as well as the roughness of the Pt-oxide and oxide-water interfaces with increasing potential. A full overview of the fit parameter values is provided in Fig. S8. The inset shows a comparison of the amount of oxide as determined via XRR (red), RDE-SXRD (blue), and the oxide reduction charge (gray) determined by Conway and Jerkiewicz[35]. Potentials are corrected for the full ohmic resistance of the electrolyte ($150 \pm 10\,\Omega$ and $33 \pm 2\,\Omega$ for the RDE-SXRD and XRR experiments, respectively, see "Methods"). Source data are provided as a Source data file.

With further increasing potential ($\geq 1.8$ V), the $Pt_{PE}$ sites show only minimal changes in occupancy while also displacement distribution $u$ increases. Also, the $Pt_{base}$ layer, now the topmost metallic layer, is gradually becoming more disordered (increasing displacement amplitude), but its occupancy only starts to decrease at the highest potentials. Interestingly, under these conditions, the in-plane position of the $Pt_{PE}$ atoms remains unaltered (including in-plane displacements does not improve the fit). Finally, in the fourth potential regime above

2 V, the $Pt_{base}$ layer gets attacked by the oxidation process and its occupancy gets reduced. This indicates that only under these conditions we start to observe bulk Pt oxidation.

To characterize the Pt oxide layer, we employed ex situ XRR, which is sensitive to the total electron density distribution perpendicular to the surface, independent from the crystalline state of the layer. It is important to note that the electrochemically formed Pt oxide is stable in air, as the XRR signal did not change over time. Under ultrahigh vacuum, on the other hand, the oxide slowly decomposes (see Fig. S11), because it is thermodynamically highly unstable[43].

Figure 3A shows XRR data (gray) that were measured after applying potentials in the regime where 2D to 3D oxide growth takes place, as determined from the RDE-SXRD experiment, i.e., 1.5–2.11 V. The appearance of weak oscillations indicates indeed the formation of an ultrathin, well-defined oxide layer. The colored lines are fits to the data using an electron density profile consisting of a Pt bulk with an oxide overlayer and a thin water layer. Ultrathin water layers may be present on oxide surfaces under ambient conditions, improving the fit quality significantly[44]. A complete overview of the fit parameter values is provided in Fig. S8. Figure 3B provides the electron density profiles perpendicular to the electrode surface during the evolution of the oxide growth, showing the metal-oxide interface moving into the Pt bulk. With increasing thickness, the oxide is also becoming slightly rougher, as can be seen from the density distribution across the interfaces becoming smeared out. The oxide density is slightly lower (−5 to −10%) than the bulk α-PtO₂ value, as indicated by the horizontal lines. The electron density is thus inline with a strongly defective α-PtO₂ layer, matching the observed lack in long-range order observed in the diffraction experiments.

The evolution of the oxide thickness as a function of potential is shown in the inset of Fig. 3B (red). Using the density determined from the XRR experiment, the total number of displaced atoms from the RDE-SXRD experiment ($Pt_{PE} + Pt_{ox}$) can be converted to an oxide layer thickness (blue), assuming $PtO_2$ stoichiometry. In this conversion, the oxidation of 1 ML of metallic Pt leads to the formation of a 4.8 Å thick platinum oxide. This approach works rather well until the first surface layer is completely oxidized (around 1.6 V). Above this value, the determination of the number of oxidized atoms from the CTR analysis get less accurate, due to the increasing number of atoms that does not contribute to the CTR intensity. As such, the SXRD data in Fig. 3B should be considered as the lower limit of the actual oxide thickness.

Overall, both RDE-SXRD and XRR data clearly show a linear increase in oxide thickness with increasing potential with a slope of $7.0 \pm 0.2$ Å V$^{-1}$. This linear trend was also reported from the (in situ) electrochemical quantification of the amount of oxide formed via its reduction[35]. Actually, when an oxide layer thickness is calculated based on the oxide reduction charge relative to 1.0 V[35], the XRR density, and assuming a Pt$^{4+}$ oxidation state, these data are in perfect quantitative agreement (see gray points in the inset of Fig. 3B).

The linear potential dependence, as well as a logarithmic time dependence of electrochemical oxide growth, are well-described in literature[9,17,35,45]. As proposed in ref. 35, the origin of this behavior is that the PE process is driven by the electric field (as confirmed by the absence of Pt oxidation during ORR, see above and Fig. S3). The formation of each $Pt_{PE}$ (and probably also $Pt_{ox}$) site may lead to a reversal of the Pt-O dipole and thereby a decrease of the effective local potential[45]. This also may lead to a repulsive interaction between individual $Pt_{PE}$ sites[46], which could explain why oxide chains/stripes that form during thermal oxidation[39,47] do not form under electrochemical conditions, at least for low $Pt_{PE}$ occupancies. In the entire potential range studied here, the sample mostly maintains it conductivity, due to the thin and defective nature of the oxide layer, making the PE field effect the determining factor for the oxide growth. This is confirmed via electrochemical impedance spectroscopy measurements, which do not show significant differences after

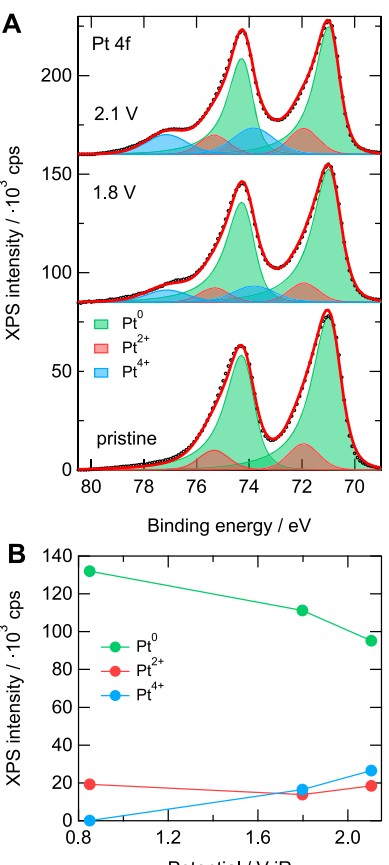

**Fig. 4 | XPS spectra and integrated intensities. A** Pt 4$f$ XPS spectra of Pt(111) after electrolyte contact at 0.8 V ("pristine") and after oxidation at 1.8 and 2.1 V. A Shirley background was subtracted from the data. **B** Evolution of the integrated intensities of the different Pt species as a function of potential. Corresponding O 1$s$ and C 1$s$ spectra are shown in Fig. S10. Potentials are corrected for the full ohmic resistance of the electrolyte ($33 \pm 2\,\Omega$, see "Methods"). Source data are provided as a Source data file.

oxidation at up to 2.1 V compared to a potential hold at 1.3 V (see Fig. S9).

If the effective local potential decreases, one would also expect to see a decrease in the OER activity, e.g., by an increase in the Tafel slope with increasing potential. Although this has been described mathematically by Damjanovic et al., this effect seems too small to quantify reliably experimentally during a potential sweep experiment[48,49]. However, potential hold experiments did indeed demonstrate a decreasing OER activity for Pt(111) after potential holds above 1.4 V[50]. A difference in the oxide thickness is also the expected origin for the decreased OER activity of Pt nanoparticles compared to bulk electrodes[51].

Both the RDE-SXRD as well as the XRR results indicate that the formed oxide exhibits characteristics of $PtO_2$, but do not provide direct chemical sensitivity. To obtain such direct information on the Pt oxidation state and thus oxide phase, we performed ex situ X-ray photoemission spectroscopy (XPS) measurements on the pristine surface and after oxidizing it at 1.8 and 2.1 V. The Pt 4$f$ spectra resulting from these measurements are shown in Fig. 4A. Similar to the approach described in ref. 52, the spectra were fitted using doublet components for Pt$^0$, Pt$^{2+}$, and Pt$^{4+}$ (using 4$f_{7/2}$ binding energies of 70.9, 71.9, and 73.8 eV, respectively) species. The integrated intensities of these different components are shown in Fig. 4B. For the pristine surface, we observe mainly Pt$^0$ species and only a small amount of Pt$^{2+}$ species. The latter is related to the exposure to the electrolyte (under

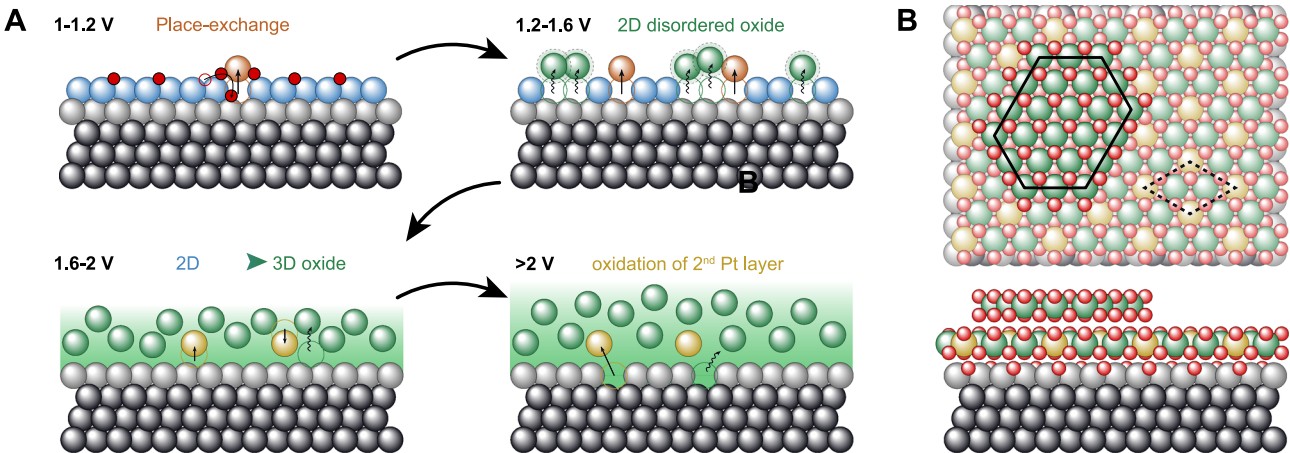

**Fig. 5 | Surface evolution. A** Schematic view of the Pt(111) oxidation process starting from the place-exchange process (top-left) until bulk oxidation (bottom-right), with $Pt_{base}$ (light gray), $Pt_{sur}$ (blue), $Pt_{PE}$ (orange, collapsed in yellow), and $Pt_{ox}$ (green). Note that the position of the $Pt_{ox}$ atoms is not known, but they are shown nonetheless to illustrate the mass conservation and increasing oxide thickness. **B** Surface structure of a commensurate α-$PtO_2$ overlayer formed by the complete oxidation of $Pt_{sur}$ based on our experimental data and literature[43]. Note that, as indicated by the black hexagon in the top view, a partial bilayer oxide is formed due to the decreased density upon oxidation.

open circuit conditions the Pt(111) surface is expected to be covered in hydroxyl species) as only metallic contributions were observed after exposing a sample prepared in UHV to air (see Fig. S12) Upon oxidizing the surface, a clear shoulder develops at higher binding energies as was previously observed for polycrystalline and nanoparticle samples[52–54], which is getting more pronounced for higher potentials. This component indicates the formation of a $Pt^{4+}$ species at the expense of the $Pt^0$ species, which further corroborates the $PtO_2$ formation found in the XRR experiments. Simultaneously, we also observe a small increase in the amount of $Pt^{2+}$ species, for which one can think of several explanations; (1) the disordered nature of the oxide leads to under-coordinated $Pt^{2+}$ atoms, (2) the electrochemically formed oxide is partially hydroxylated[55]; or (3) the formation of a PtO interface layer[42]. Importantly, these descriptions are not fully mutually exclusive, e.g., the Pt atoms on defective sites could bind hydroxyl groups. If an oxygen double layer is present at the metal-oxide interface (see above), interfacial Pt atoms could be in a $Pt^{2+}$ oxidation state. However, considering that the formed oxide layer is extremely thin, combined with its relative instability in UHV (as is the thermal oxide[56], see also Fig. S11), in situ spectroscopic measurements might be required to resolve such details.

To summarize, Fig. 5A shows the overall electro-oxidation scenario, which is dominated up to around 1.2 V by the Pt atom extraction and place-exchange process (top-left). In the potential regime from 1.2 to 1.6 V, surface atoms are displaced into positions that are not in registry with the underlying lattice anymore (top-right). Although this structure is disordered, the fact that a significant fraction of $Pt_{ox}$ returns to its original lattice position upon reduction[9], suggests that these atoms do not get displaced over very large distances. They could, for example, be involved in the formation of reconstructed oxide stripes, although large domains of such structures can be excluded. Around 1.6 V, the first restructuring of the oxide layer towards bulk $PtO_2$ takes place. Without the lateral interaction with the $Pt_{sur}$ layer, we find the atoms at the former $Pt_{PE}$ positions much closer to the $Pt_{base}$ layer, without changing their in-plane position. These atoms are now part of the growing oxide layer, but possess a well-defined position with respect to the Pt(111) surface, such that they contribute to the CTR diffraction signal. 3D oxide growth is only clearly observed at potentials around 2.0 V, at which the next Pt layer, $Pt_{base}$, gets attacked (bottom-right). The now topmost metal layer, $Pt_{base}$, starts to get displaced as seen from an increase in the static displacement distribution $u$ and a reduction in occupancy of this layer. Within

the entire potential window studied here, the total oxide thickness measured by XRR increases linearly with the applied potential. This can be explained by the high electric field across the growing oxide layer, which is of the order of 1 V $nm^{-1}$ in the case of electrochemical oxidation, enhancing field-induced ion diffusion significantly[57]. For comparison, in thermal oxidation, such fields are typically below 0.1 V $nm^{-1}$, arising from electron transfer from the metal at the interface to the chemisorbed oxygen at the surface of the growing oxide layer[58]. The smaller electric fields lead to a larger contribution of thermally excited ion currents during the oxidation process. Furthermore, its composition, i.e., $PtO_2$, likely with a double O layer at the oxide/metal interface, remains constant. Combining the data from the RDE-SXRD, XRR, and XPS experiments, it is possible to describe the local oxide structure based on the commensurate α-$PtO_2$/(2 × 2)Pt(111) interface as described in ref. 43. This is illustrated in Fig. 5B for the situation that occurs when the complete $Pt_{sur}$ layer is oxidized (around 1.6 V). Due to the decreased (Pt atom) density of the oxide, this leads to the formation of a partial bilayer oxide. The vertical displacements are based on those of the $Pt_{PE}$ atom positions and the $PtO_2$ bulk lattice constant for the first and second layer, respectively. This structure has an average oxide thickness of 4.5 Å with a surface roughness of 1.8 Å, which matches well with the XRR results (see Figs. 3B and S8B). Due to the sluggish oxidation kinetics of Pt at room temperature, this structure can only form locally without long-range order, as that would lead to surface rod features in the RDE-SXRD experiment.

Resolving electrode-electrolyte interfaces under operando electrochemical conditions is experimentally challenging, but provides crucial insights into the structural and chemical properties of the active phase of electrocatalysts. Here, we used both diffraction and spectroscopy-based X-ray techniques to deepen our understanding of the electro-oxidation of Pt(111) during the oxygen reduction and OERs at potentials up to 2.1 V, hitherto unexplored. For this, we used a novel approach combining RDE and HE-SXRD, allowing us to study the electrode/electrolyte interface at steady state OER currents much closer to electrolyzer applications compared to previous studies. We observed that the metallic surface is affected by the oxidation process in a layer-by-layer manner, the extent of which is solely determined by the electrochemical potential. This self-limiting oxide growth, originating from a local electric dipole effect, explains the high stability of Pt electrodes under oxidizing conditions (especially at constant potential). For example, it is much less destructive than a kinetic roughening during the electro-oxidation process, which may lead to

void formation at the interface and blistering of the oxide, resulting in fast electrode degeneration[59]. The local electric dipole effect is also the reason that Pt(111) remains unaffected under ORR conditions, i.e., in the presence of $O_2$, but in the absence of a strong enough electric field. Nonetheless, it should be noted that the ORR is affected by the (oxidation-reduction) history of the sample[60]. Finally, we obtained an improved understanding in the differences between electrochemical and thermal oxidation, the different driving force makes the latter explicitly not self-limited. This results in a distinctly different bulk oxidation behavior, even though the initial, surface oxidation process appears quite similar. A comparable protective oxide layer is expected to form on other Pt basal planes, e.g., Pt(100)[35], which exhibit a uniform interfacial electric field. However, the presence of step/particle edges introduces a variation on the local electric field, which likely leads to enhanced/accelerated oxidation. Therefore, to fully understand the role of the oxide layer with respect to both surface passivation as well as corrosion protection in industrial applications, further experiments on less densely-packed surfaces should be explored.

Our findings provide valuable information for theoretical simulations of the platinum(oxide)-electrolyte interface. Despite the increase in disorder compared to the pristine surface, we have been able to set important boundaries regarding the amount of material involved in the oxide formation as well as pinpoint some crucial atomic positions within the oxide structure. This drastically limits the parameter space that has to be screened in a theoretical study. Such work, as well as in situ spectroscopic experiments, would help us to not only get a detailed understanding the oxide structure itself, but in particular its interaction with the electrolyte species. This information would not only be relevant for the Pt model system, but also for OER electrocatalysts (which are typically oxidic in nature) in general. set the experimental ground for future theoretical simulations of this complex interface under OER conditions.

## Methods

Prior to the experiments, all glassware and parts with direct contact to the electrolyte were soaked in a 1 g/L $KMnO_4$/0.5 M $H_2SO_4$ solution for at least 24 h and subsequently boiled at least five times in ultrapure water (>18.2 MΩ cm). Where applicable, the syringe pump and all electrolyte-carrying tubes were flushed extensively with hot ultrapure water. The hat-shaped Pt(111) sample (cut and polished <0.1°, MaTeck) with a 4.6 mm diameter circular exposed area was prepared by flame annealing (5 min at ~1250 K), and cooling in a reducing atmosphere (1:4 $H_2$/Ar mixture). Bromine adsorption was used to protect the surface against contamination during the RDE prealignment[31,61]. High-purity perchloric acid (99.999% trace metals basis, Sigma Aldrich) was used to prepare the 0.1 M $HClO_4$ electrolyte solution. Several hours prior to and during the experiment, the electrolyte was purged with Ar or $O_2$. All experiments were performed in a three-electrode configuration, using a miniature reversible hydrogen electrode (RHE, gaskatel) or Ag/AgCl electrode (eDAQ, ET072) as reference electrode, and a coiled platinum wire (99.999%, MaTeck) as counter electrode. A potentiostat (PGSTAT204, Metrohm) was used to control the potentials and collect the electrochemical data. Prior to the experiments, reference electrodes are calibrated against a dedicated calibration RHE in the working solution. Potentials are reported versus the reversible hydrogen electrode scale. Corrections for the full ohmic resistance of the electrolyte (150 ± 10 and 33 ± 2 Ω in the RDE-SXRD and glass cell experiments, respectively) were applied during data processing for all data except the cyclic voltammetry shown in Fig. S2A.

The instrument for combined RDE and HE-SXRD experiments is shown in Fig. S1 and has been described extensively in previous work[31]. Briefly, the setup enables HE-SXRD experiments in a hanging meniscus geometry while the sample rotates fast around its surface normal. The glass cell, containing approximately 250 mL of the electrolyte solution,

can be translated vertically to control the meniscus without affecting the sample alignment. For the experiments reported here, the rotation speed was set to 600 rpm. The operando SXRD experiments were performed at the second experimental hutch (EH2) at beamline P07 at the PETRA III light source at the Deutsches Elektronen Synchrotron (DESY), Hamburg[62]. Lead absorber pieces were used to block the high-intensity Bragg peaks, leading to gaps in the data at those positions. Note that the continuous RDE rotation across 360 degrees required all Bragg peaks to be blocked, which, due to the sample symmetry, leads to gaps in the data, e.g., at $L = 1$ in the (01) CTR. The incidence angle was set at 0.03 degrees, close to the angle for external total reflection for Pt at the employed photon energy of 74.8 keV, to improve the signal-to-noise ratio. Diffraction patterns were recorded with a high-energy 2D area detector (Pilatus3X CdTe 2M, Dectris), with a total sensitive area of 253.7 × 288.8 mm² and a pixel size of 172 × 172 μm², which was placed at a distance of 800 mm from the sample. A hexagonal surface unit cell ($a = b = 2.77$ Å, $c = 6.80$ Å) was chosen such that the $a^*$ and $b^*$ reciprocal space vectors describe the surface plane while the $c^*$ vector lies along the surface normal. The corresponding H, K, and L coordinates were defined using a Pt lattice constant of 3.924 Å. Data processing[63], i.e., background subtraction, intensity correction, and signal integration was performed using home-written scripts in Wavemetrics Igor Pro. The ANAROD package was used to fit the structural models to the data[64]. Including an overall scale factor, a total number of twelve fit parameters (occupancies, displacements, and Debye–Waller factors, see main text) is used. The bulk Debye–Waller factor is 0.38 as tabulated in literature[65]. Although quantitative values determined from the synchrotron experiments are based on individual measurements, (partial) repeats were performed to ensure reproducibility. Error bars indicate the errors of the least-squares fit.

Prior to the ex situ XRR and XPS measurements, the sample was characterized and oxidized in a glass cell (80 mL electrolyte volume) using a hanging meniscus configuration. For the oxidation, a slow potential sweep (1 mV s⁻¹) was used, followed by a 30 min. potential hold at the upper potential limit to ensure equilibration of the surface structure. To increase the meniscus quality/stability during these measurements, they were performed under static conditions. This is not expected to influence the result of the measurement as the oxidation process is insensitive to the presence of $O_2$. Afterwards, the meniscus was broken under potential control, and the sample was rinsed with ultrapure water. XRR measurements were performed in a $\theta/2\theta$ geometry using Cu $K_\alpha$ radiation ($\lambda = 1.54051$ Å) at DESY NanoLab[66]. The data were analyzed using Fewlay[67]. All XPS measurements were carried out at the DESY Nanolab at the Center for X-ray and Nano Science, DESY, Hamburg[66]. The instrument is equipped with a monochromated Al $K_\alpha$ source (1.486 keV) and a Phoibos 150 hemispherical energy analyser, operating under a base pressure of low 10⁻¹⁰ mbar. The X-ray source was operated at 15 kV and 20 mA, corresponding to a total power of 300 W. Pt $4f$ and O $1s$ core levels are measured at an energy pass of 20 with a step size of 0.1 eV. Data analysis was performed using CasaXPS software[68]. All XP peaks have been fitted using a Shirley-type background and a Gaussian–Lorentzian line shape. For the fitting of the Pt $4f$ peaks, a fixed spin-orbit coupling of 3.3 eV, separating the $4f_{7/2}$ and $4f_{5/2}$ peaks, was used. The area ratio of the $4f_{7/2}$ and $4f_{5/2}$ contributions was, respectively, fixed at 4:3. The full width at half maximum was kept consistent between spin-orbit components. Fitting parameters are reported in Table S1.

## Data availability

Data are available as source data file in the Supplementary Material. Source data are provided with this paper.

## Code availability

All software used for this study is publicly available or can be obtained from the corresponding author upon reasonable request.

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

## Acknowledgements

We acknowledge DESY (Hamburg, Germany), a member of the Helmholtz Association HGF, for the provision of experimental facilities. Parts of this research were carried out at PETRA III, and beamtime was allocated for proposals I-20190341 and I-20200593 (L.J., V.V., and A.S.). We thank Simon Geile for assistance in constructing the RDE-SXRD setup. This work was financially supported by the German Federal Ministry of Education and Research (BMBF projects 05K2016-HEXCHEM (H.O., A.S.) and 05K19WE1-CIXenergy (J.L., A.S.)). The authors acknowledge additional support by the Bavarian Ministry of Economic Affairs, Regional Development and Energy, and the Deutsche Forschungsgemeinschaft (DFG) via the Collaborative Research Center SFB 1452 (Catalysis at Liquid Interfaces, project 431791331), the Research Unit FOR 1878 (Functional Molecular Structures on Complex Oxide Surfaces, project 214951840), and further projects (431733372, 453560721) (J.L.).

## Author contributions

L.J.: RDE-SXRD, XRR, XPS, analysis, and manuscript preparation; R.S.: RDE-SXRD, analysis, and manuscript preparation; M.K.: XPS, analysis; D.S.D.: RDE-SXRD, XRR, and analysis; J.P.: RDE-SXRD; XD: RDE-SXRD; T.W.: RDE-SXRD; O.G.: RDE-SXRD; A.-C.D.: RDE-SXRD; O.B.: RDE-SXRD; Y.L.: RDE-SXRD, XPS; H.N.: XPS; H.O.: RDE-SXRD; J.L.: RDE-SXRD; V.V.: RDE-SXRD, XRR; A.S.: RDE-SXRD, XRR, and XPS.

## Funding

## Competing interests

The authors declare no competing interests.

## Additional information

**Supplementary information** The online version contains Supplementary material available at https://doi.org/10.1038/s41467-026-72954-z.

