## [Transparent Peer Review File · Nature Communications]

Platinum Oxide Formation under Oxygen Evolution Reaction Conditions

Corresponding Author: Dr Leon Jacobse

Version 0:

Reviewer comments:

Reviewer #2

(Remarks to the Author)

I still believe that this manuscript is very good, generally. Furthermore, the authors have addressed most of the raised points appropriately. However, there is an important point that has not been resolved.

“All XP peaks have been fitted using a Shirley background and a Gaussian-Lorentzian line shape.” Pedantically, one could argue that the employed fit function is still ambiguously described, as it can relate to three different (but very similar) functions. However, crucially, these are all symmetric functions. Looking at Fig 4, Pt del + and Pt0 have been fitted with asymmetric functions.

The reason this is important is that all the asymmetric functions used in XPS fitting are highly empirical, and therefore prone to bias. The presence of the delta + peak in the Pt4f is still far from convincing. This delta plus peak is a rather large contribution (larger than the 2 and 4+ for most spectra.), so its justification needs to be solid.

Closely examining the fitted spectra, it seems that the 1.8 V and 2.1 V spectra are not very well described by the total fit. This is most noticeable in the top spectrum between 76-77 eV. It does not look too bad, but considering there are about 18 fitting parameters, it is not great.

“Although a distinct shoulder is not visually apparent, the improved fit quality upon inclusion of this component, together with precedent in the literature under similar electrochemical conditions, supports the presence of an intermediate oxidation state likely associated with partially oxidized or sub-stoichiometric platinum species.”

Fits always improve by adding more fitting parameters. This is not a prove that there is any physical justification for adding more fitted species. Equally, citing literature is not necessarily helpful as many examples of poorly fitted XPS data have been published.

In my opinion, more rigorous prove is needed for the existence of the delta plus peak. Without this, I do not believe the XPS results can be published.

1. Using the same fitting model, fit the Pt4f of a sputter-cleaned Pt(111) single crystal, showing that there is no need for the delta plus peak to obtain sufficiently good fit results
2. Show the fits in Fig 4 with the intensity of the delta plus peak constrained at 0.
3. Fit the spectra in Fig 4 using different asymmetric functions: Doniach-Sunjic, Mahan, etc.
4. If points 1-3 show that this peak is indeed real, the relevance for the experiments need to be shown by taking the cleaned, pristine Pt(111) and expose it to air for a few hours before remeasuring it. This would eliminate the possibility that this peak is an artefact of the air transfer.

Reviewer #3

(Remarks to the Author)

The reviewer appreciates the efforts the authors devoted in revising the manuscript. While the revised manuscript has addressed many of the concerns raised before, the following entries need further consideration.

“5. Have the authors detected dissolved Pt in the electrolyte using ICP analysis after the experiments? Unfortunately, we have not had the opportunity to perform such measurements, which is why we rely on literature data which is available for the lower potentials. Under those conditions, the amount of dissolved platinum is far below the uncertainty of the SXRD measurements.”

ICP should be attempted to determine the rate of Pt dissolution, as it is crucial to understand a key potential effect of Pt oxidation.

“8. The current version of the paper lacks discussion on the correlation between surface structure and the OER activity, which is expected to a top catalysis journal.

We acknowledge this argument and realize that the gradually increasing oxide thickness, without clear trend changes in the OER activity, provides limited insights in the effect of the surface (oxide) structure on OER activity. This is, however, not the main point of our study.”

This current study is motivated by the implication of enhancing OER with the mechanistic understanding achieved by the advanced characterization results (see abstract and introduction). Then, the implication of the oxidation mechanism on the reaction should be an organic part of this work.

Reviewer #5

(Remarks to the Author)

This paper mainly investigates the structural evolution and oxidation mechanism of the Pt(111) electrode under high potentials by means of in situ high-energy X-ray diffraction (HE-SXRD), X-ray reflectivity (XRR), and X-ray photoelectron spectroscopy (XPS). The authors conclude that electrochemical oxidation is driven by the interfacial electric field rather than by thermodynamics, leading to a “self-limiting” behavior that produces a sub-nanometer-thick, defect-rich PtO₂ layer protecting the underlying Pt substrate. This mechanism differs fundamentally from thermal oxidation. Understanding such an electric-field-driven, layer-by-layer oxidation process is valuable for developing more stable OER catalysts. The experimental design is well executed and the characterization is clear, so the manuscript can be considered for acceptance after the following issues are addressed:

1. In the Introduction, please provide a concise summary of existing in situ characterization studies on Pt and related metals, so as to better clarify the significance and novelty of the present work.
2. Why was the study performed in 0.1 M HClO₄? Would the pH of the electrolyte influence the oxidation behavior and the obtained results?
3. The applied potential range is 0–2 V. During the ORR process, Pt is reduced, and during OER, Pt is not the main active catalyst. How can the present findings be correlated with real reaction conditions? Please expand the discussion.
4. The authors chose the (111) facet for investigation. Would other facets exhibit a similar potential-dependent thickness behavior, and how might this affect the activity and stability toward OER or ORR?
5. Does the formation of the surface PtO₂ layer alter the reaction pathways of ORR and OER?
6. The current work relies solely on spectroscopic techniques. Could morphological characterizations (e.g., ex situ AFM or TEM) be provided to corroborate the spectroscopic results?

Version 1:

Reviewer comments:

Reviewer #2

(Remarks to the Author)

The additional experiments performed by the authors have addressed all the points I raised. I believe the manuscript has improved significantly and is ready to be accepted for publication in the Nature Communication.

Reviewer #3

(Remarks to the Author)

Overall, the manuscript is fit for publication as it has addressed the main concerns of mine. I have to say it is a bit curious that the authors refused to do a simple ICP test of the electrolyte to determine the level of Pt dissolution. I don't consider it to be a routine, rather than an excessive, request.

Reviewer #5

(Remarks to the Author)

Now I think this paper should be published at present form.

Reviewer #2

1. Using the same fitting model, fit the Pt4f of a sputter-cleaned Pt(111) single crystal, showing that there is no need for the delta plus peak to obtain sufficiently good fit results.

We thank the reviewer for this valuable suggestion. We have applied the model recommended by the reviewer. In addition, we performed an independent check to identify the presence of oxidized Pt on the pristine Pt(111) surface after cleaning by Ar⁺ sputtering, in order to compare the sputter-annealed surface with the commonly used flame-annealed preparation.

The Pt(111) crystal was cleaned by four cycles of Ar⁺ ion sputtering (1.5 kV, 15 min) followed by annealing to 1000 K in UHV for 3 minutes. The surface cleanliness and order were verified by LEED and XPS, as shown in Figures S11a and S11b. XPS measurements were performed using an Al K α source (1.486 keV, 400 W). Analysis of the XPS spectra was performed using the CasaXPS software. A Shirley background subtraction was used in all cases. The asymmetric Lorentzian LF line shape was used for fitting of the Pt 4f spectra (LF(0.9, 2, 20, 50)), which is widely used as a practical asymmetric approximation for metallic Pt surfaces [1].

For both the sputter-annealed surface and the surface after brief exposure to air (2 hours), only sharp metallic Pt 4f signals were observed (Figures S11c and S11d). Using the LF line shape, the FWHM of metallic Pt components was 1.1 eV.

Figure S11: (a) LEED pattern (at 85 eV) and (b) XPS survey scan of the clean Pt(111) surface after sputter-anneal preparation. Deconvoluted Pt 4f core-level spectra of the sample (c) after sputtering-annealing and (d) after 2 hours of exposure to air.

[1] R. Mom, et al., The Oxidation of Platinum under Wet Conditions Observed by Electrochemical X-ray Photoelectron Spectroscopy. *J. Am. Chem. Soc.*, **2019**, 141(16), 6537-6544. <https://doi.org/10.1021/jacs.8b12284>

2. Show the fits in Fig 4 with the intensity of the delta plus peak constrained at 0.

As described in our response to comment 1, we re-fitted the Pt 4f spectra obtained after flame annealing and after applying electrochemical potentials using the fitting parameters reported in Ref. 1. The updated fits and corresponding Pt 4f spectra are now presented in the revised Figure 4 of the manuscript.

3. Fit the spectra in Fig 4 using different asymmetric functions: Doniach-Sunjic, Mahan, etc.

We have re-fitted the data according Reviewer’s suggestion. Based on this updated analysis, the previously assigned $Pt^{\delta+}$ component is no longer present. Only Pt^{2+} and Pt^{4+} species are observed in the new fitting results. Fitting parameters are reported in table S1.

	Pt^0	Pt^{2+}	Pt^{4+} (Determined at 1.8 and 2.1 V)
Line shape	LF(0.9, 2, 20, 50)	GL(30)	GL(30)
Peak Position	70.9, 74.2 eV	71.9, 75.3	73.8, 77.1
FWHM	0.96-1.1 eV	1.3 eV	1.8 eV

4. If points 1-3 show that this peak is indeed real, the relevance for the experiments need to be shown by taking the cleaned, pristine Pt(111) and expose it to air for a few hours before remeasuring it. This would eliminate the possibility that this peak is an artefact of the air transfer.

Thank you again for the comment. This point has already been addressed in our responses to comments 1-3, where we updated our fitting model according to the reviewer's suggestion.

Reviewer #3 (Remarks to the Author):

The reviewer appreciates the efforts the authors devoted in revising the manuscript. While the revised manuscript has addressed many of the concerns raised before, the following entries need further consideration.

“5. Have the authors detected dissolved Pt in the electrolyte using ICP analysis after the experiments? Unfortunately, we have not had the opportunity to perform such measurements, which is why we rely on literature data which is available for the lower potentials. Under those conditions, the amount of dissolved platinum is far below the uncertainty of the SXRD measurements.”

ICP should be attempted to determine the rate of Pt dissolution, as it is crucial to understand a key potential effect of Pt oxidation.

We acknowledge that noble metal dissolution under OER conditions is a relevant topic and cannot simply be ignored. However, based on the available literature described below, we disagree with the reviewer that additional dissolution experiments are crucial to validate our work on oxide formation on Pt(111). Cherevko et al. described the dissolution of Pt(poly) and other noble metals during potential cycling in H₂SO₄ (DOI: [10.1002/cctc.201402194](https://doi.org/10.1002/cctc.201402194)), which shows that all noble metals dissolve under OER conditions. It was shown that only Pd shows a lower dissolution rate than Pt, and Pt dissolution is almost exclusively a transient process, rather than steady-state, with the majority of the dissolution occurring during the oxide reduction. Follow-up studies by Sandbeck et al. (DOI: [10.1002/cphc.201900866](https://doi.org/10.1002/cphc.201900866)), Briega-Martos et al. ([10.1002/celec.202300554](https://doi.org/10.1002/celec.202300554)), and Fuchs et al. ([10.1002/anie.202304293](https://doi.org/10.1002/anie.202304293)) demonstrated that the Pt(111) surface orientation is the most resistant to dissolution. These studies confirmed, once again, that anodic dissolution is mostly a transient process, becoming largely time- and potential-independent above 1.4 V_{RHE}. Most importantly for our work, is the fact that the total amount of dissolved Pt is always significantly below 1% of a monolayer, far beyond the sensitivity of the SXRD experiment. Considering this extremely low dissolution rate, one might even wonder if dissolution takes place at Pt(111) terraces at all, or only at naturally-occurring defects, which is however beyond the topic of the current study. For clarification, we have made our statement regarding dissolution in the main text more explicit:

We use the net reduction in occupancies of these observable layers (Pt_{base} , Pt_{sur} , Pt_{PE}) to determine the upper limit for the amount of disordered oxide (Pt_{ox}), considering dissolution negligible under the applied experimental conditions.²²

→

Realizing that Pt dissolution under our applied experimental conditions is far below the sensitivity of the SXRD experiment (<<1% of a ML), we use the net reduction in occupancies of these observable layers (Pt_{base} , Pt_{sur} , Pt_{PE}) to determine the upper limit for the amount of disordered oxide (Pt_{ox}).²²⁻²³

“8. The current version of the paper lacks discussion on the correlation between surface structure and the OER activity, which is expected to a top catalysis journal. We acknowledge this argument and realize that the gradually increasing oxide thickness, without clear trend changes in the OER activity, provides limited insights in the effect of the surface (oxide) structure on OER activity. This is, however, not the

main point of our study.”

This current study is motivated by the implication of enhancing OER with the mechanistic understanding achieved by the advanced characterization results (see abstract and introduction). Then, the implication of the oxidation mechanism on the reaction should be an organic part of this work.

We have incorporated several changes throughout the manuscript, also based on the comments from Reviewer #5 to strengthen the connection between the observed structural changes and the electrocatalytic OER activity.

Reviewer #5 (Remarks to the Author):

This paper mainly investigates the structural evolution and oxidation mechanism of the Pt(111) electrode under high potentials by means of in situ high-energy X-ray diffraction (HE-SXRD), X-ray reflectivity (XRR), and X-ray photoelectron spectroscopy (XPS). The authors conclude that electrochemical oxidation is driven by the interfacial electric field rather than by thermodynamics, leading to a “self-limiting” behavior that produces a sub-nanometer-thick, defect-rich PtO₂ layer protecting the underlying Pt substrate. This mechanism differs fundamentally from thermal oxidation. Understanding such an electric-field-driven, layer-by-layer oxidation process is valuable for developing more stable OER catalysts. The experimental design is well executed and the characterization is clear, so the manuscript can be considered for acceptance after the following issues are addressed:

1. In the Introduction, please provide a concise summary of existing in situ characterization studies on Pt and related metals, so as to better clarify the significance and novelty of the present work.

Several citations were added throughout the introduction, discussion, and conclusions.

2. Why was the study performed in 0.1 M HClO₄? Would the pH of the electrolyte influence the oxidation behavior and the obtained results?

Perchloric acid is the typical electrolyte used for these experiments because its interaction with the Pt(111) surface is negligible. In the presence of stronger interacting species the oxidation process is probably also affected. Identifying an independent pH effect is more complicated. A sentence on the choice of the electrolyte was added to the main text.

To avoid effects originating from the interaction between (both anionic and cationic) electrolyte species, we use a 0.1 M HClO₄ electrolyte solution.

3. The applied potential range is 0–2 V. During the ORR process, Pt is reduced, and during OER, Pt is not the main active catalyst. How can the present findings be correlated with real reaction conditions?

Please expand the discussion.

Indeed, we observe no oxide formation under ORR conditions although oxygenated species must be present at the surface as confirmed by our ex situ XPS measurement. Nonetheless, Pt catalysts can be exposed to potentials up to 2 V during shutdown of a PEM fuel cell as referenced in the introduction. The extent of the catalyst oxidation (and subsequent reduction when operation is resumed) will strongly affect the lifetime of the catalyst.

4. The authors chose the (111) facet for investigation. Would other facets exhibit a similar potential-dependent thickness behavior, and how might this affect the activity and stability toward OER or ORR?

We expect similar observations, albeit maybe with a different slope, for other basal planes, which exhibit a uniform electric field perpendicular to the surface plane. The behavior of stepped surfaces or nanoparticles might be more complicated as the local interfacial electric field will be different at undercoordinated sites (Smoluchowski effect). We have elaborated on this in the conclusions:

A comparable protective oxide layer is expected to form on other Pt basal planes, e.g. Pt(100)³³, which exhibit a uniform interfacial electric field. However, the presence of step/particle edges introduces a variation on the local electric field, which likely leads to enhanced/accelerated oxidation (and dissolution). Therefore, to fully understand the role of the oxide layer with respect to both surface passivation as well as corrosion protection in industrial applications, further experiments on less densely-packed surfaces should be explored.

5. Does the formation of the surface PtO₂ layer alter the reaction pathways of ORR and OER?

Although the PtO₂ layer is likely not stable under ORR conditions, the restructuring and formation of oxygenated adsorbates are expected to affect the ORR. A reference to recent literature has been added to the conclusions.

Nonetheless, it should be noted that the ORR is affected by the (oxidation-reduction) history of the sample.⁵⁵

Although OER does not take place on the metallic surface, the amount of PtO₂ formed does affect the activity (thicker oxide is less active). The following text was added to the discussion:

If the effective local potential decreases, one would also expect to see a decrease in the OER activity, e.g. by an increase in the Tafel slope with increasing potential. Although this has been described mathematically by Damjanovic et al., this effect seems too small to quantify reliably experimentally during a potential sweep experiment.^{47,48} However, potential hold experiments did indeed demonstrate a decreasing OER activity for Pt(111) after potential holds above 1.4 V.⁴⁹ A difference in the oxide thickness is also the expected origin for the decreased OER activity of Pt nanoparticles compared to bulk electrodes⁵⁰.

6. The current work relies solely on spectroscopic techniques. Could morphological characterizations (e.g., ex situ AFM or TEM) be provided to corroborate the spectroscopic results?

We have studied the oxide / metal interfacial structure by surface sensitive x-ray diffraction methods and in addition by spectroscopic investigations. The surface x-ray diffraction data give us atomistic insight into the structurally ordered part of the interface. X-ray reflectivity provides us with the interfacial electron density profile and with that (surface) morphological information of the oxide layer and the interface, as described in our article. Together with the XPS results, these methods give a consistent picture of the structure, phase composition and roughness at the interface while in contact with the electrolyte at relatively high (electrochemical) potentials. Although in our study we have not used AFM or TEM, these of course are also powerful methods and references to this kind of studies (21,25) have been added to the introduction.